# HI-TOM: A Benchmark for Evaluating Higher-Order Theory of Mind Reasoning in Large Language Models

**Yinghui He**[*]🍑, **Yufan Wu**[*]🍑, **Yilin Jia**🍑, **Rada Mihalcea**🍑, **Yulong Chen**🍋, **Naihao Deng**[†]🍑

🍑 University of Michigan    🍋 Westlake University
{huihui, umwyf, kripf, mihalcea, dnaihao}@umich.edu

## Abstract

Theory of Mind (ToM) is the ability to reason about one's own and others' mental states. ToM plays a critical role in the development of intelligence, language understanding, and cognitive processes. While previous work has primarily focused on first and second-order ToM, we explore higher-order ToM, which involves recursive reasoning on others' beliefs. We introduce **HI-TOM**, a **Hi**gher Order **T**heory **o**f **M**ind benchmark. Our experimental evaluation using various Large Language Models (LLMs) indicates a decline in performance on higher-order ToM tasks, demonstrating the limitations of current LLMs. We conduct a thorough analysis of different failure cases of LLMs, and share our thoughts on the implications of our findings on the future of NLP.

## 1 Introduction

Theory of Mind (ToM) refers to the ability to understand and reason about the mental states of others such as intentions and beliefs, and also to distinguish them from one's own (Premack and Woodruff, 1978). Such an ability has been considered a crucial point in the development of intelligence functions (Premack and Woodruff, 1978; Bretherton and Beeghly, 1982; Frith and Frith, 2003), and previous research has demonstrated that ToM reasoning is highly related to linguistic and cognitive processes (Perner, 1991; Sperber and Wilson, 2002). ToM has thus been widely used as a protocol to evaluate the language understanding and reasoning ability of intelligence agents (Premack and Woodruff, 1978; Takano et al., 2006), such as young children (Osterhaus and Koerber, 2021).

With the recent advance in large language models (LLMs), research has been undertaken to evaluate the language skills of LLMs using ToM (Sap et al., 2022; Ullman, 2023). Most of the previous

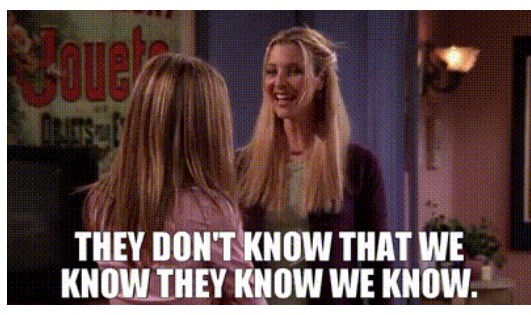

Figure 1: A scene shot from the TV series *Friends* that exhibits fourth-order Theory of Mind (ToM).

work has been confined to first-order and second-order ToM, where LLMs are asked to perform inference on others' belief of reality in one or two passes, e.g., the first and second-order questions in Figure 2 (see Section 2 for a more comprehensive discussion of ToM background and the evaluation of ToM in LLMs).

Higher-order ToM, referring to third-order reasoning and beyond, requires recursive reasoning on others' beliefs in multiple passes. Figure 1 shows a higher-order ToM example from the TV series *Friends*. In Figure 1, a character says "*They don't know that we know they know we know*" when she and the other character try to recursively identify the situation. Such an example underscores that human beings are capable of higher-order ToM in daily interactions. In addition, evidence shows that higher-order ToM is not only essential to communicate effectively in complicated scenarios, such as multi-party conversations (Liddle and Nettle, 2006; De Weerd et al., 2015; Ridinger and McBride, 2017; De Weerd et al., 2022), but it also enables better emotional support and empathetic communication (Mitchell and Phillips, 2015). However, because of a lack of higher-order ToM datasets in the NLP community, there is significantly less research on higher-order ToM compared to the lower orders.

Previous work has mainly constructed ToM

---

[*]Contributed equally to this work.
[†]Corresponding author of this work.

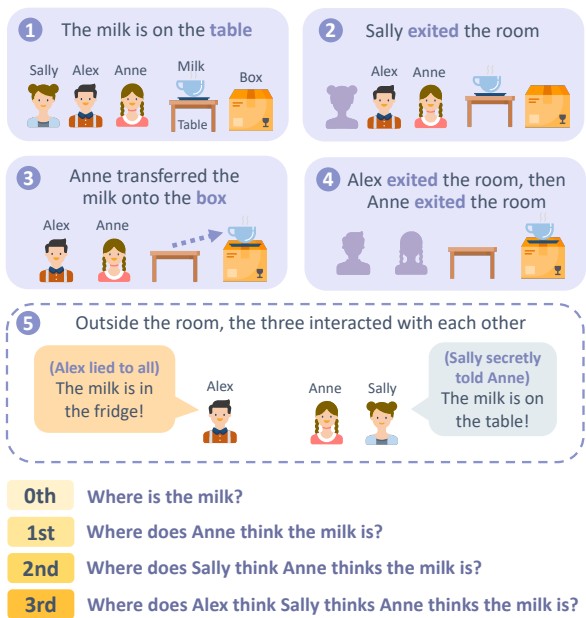

**0th** Where is the milk?

**1st** Where does Anne think the milk is?

**2nd** Where does Sally think Anne thinks the milk is?

**3rd** Where does Alex think Sally thinks Anne thinks the milk is?

Figure 2: A sample from HI-TOM dataset, which contains communications among agents, and questions that address 0-th (reality) to 3-rd ToM reasoning..

benchmarks using automatic story generation scripts. Although simple and inexpensive, this method cannot be directly extended to generating stories of higher-order ToMs because the generated stories contain insufficient information for raising a higher-order question. In this paper, we build upon previous work and introduce **HI-TOM**, a multiple-choice question benchmark consisting of Sally-Anne-like stories (Figure 2), specifically designed for higher-order ToM evaluation. Unlike previous datasets, HI-TOM contains questions from zeroth-order to fourth-order ToM, and incorporates agent communications in the stories. We manually check the quality of the constructed data, and empirically find that HI-TOM presents greater diversity and challenges compared to previous datasets.

We experiment with various LLMs, including GPT-4 (OpenAI, 2023), GPT-3.5-turbo (OpenAI, 2022), Claude, and Guanaco (Dettmers et al., 2023), on HI-TOM under a zero-shot setting. Furthermore, we test the chain-of-thought prompting (Wei et al., 2022) and conduct a thorough analysis of LLMs' performances on different story types in HI-TOM and their failure cases. Our work demonstrates that the claim of LLMs having genuine ToM abilities (Kosinski, 2023; Bubeck et al., 2023) is questionable, especially in the cases of higher-order ToM, where several rounds of recursive reasoning are required. To our knowledge, we are the first to introduce a benchmark for evaluat-

ing higher-order ToM reasoning and analyzing the abilities of current LLMs on high-order ToM. Furthermore, we share our thoughts on the future of NLP and the way forward with LLMs of enhancing LLMs from the perspective of human intelligence, understanding humans through the lens of LLMs, and enhancing LLMs' ToM abilities for better NLP applications. We release our dataset and code at `https://github.com/ying-hui-he/Hi-ToM_dataset`.

## 2 Background and Related Work

**Theory of Mind.** Most of prior work focuses on first or second-order ToM (Nematzadeh et al., 2018; Le et al., 2019; Sap et al., 2022), while higher-order ToM (third-order and beyond) remains under-explored. The concept of "*orders*" refers to the number of mental state attributions that are required to answer a particular question or reason about a particular scenario. For instance, a third-order ToM question can be "*Where does Anne think that Sally thinks that Isabella searches for the milk?*", where Sally's reasoning about Isabella is of second-order, and Anna's reasoning on Sally's reasoning is of third-order.

Higher-order ToM is useful in social interaction such as maintaining social networks (Liddle and Nettle, 2006), winning limited bidding (de Weerd and Verheij, 2011), efficiently cooperation (De Weerd et al., 2015; Ridinger and McBride, 2017), and unpredictable negotiations (De Weerd et al., 2022). Researchers from cognitive science investigate second-order and higher-order ToM among young children via complex forms of false-belief tests, such as the Sally-Anne false-belief experiment (Baron-Cohen et al., 1985).

**Evaluating ToM in LLMs.** Sap et al. (2022) find that GPT-3's ToM ability is well below humans on the TOMI dataset (Le et al., 2019), which is a ToM evaluation dataset consisting of questions up to the second order. Kosinski (2023); Bubeck et al. (2023) show the promising performance of recent LLMs such as GPT-3.5 and GPT-4 on ToM tasks. However, it is questionable whether LLMs have genuine ToM ability, especially for higher-order ToM. Ullman (2023) find that for GPT-3.5, small variations that maintain the principles of ToM can cause a flip of the answer. Different from previous work that only evaluates LLMs' ToM ability up to the second order, we take a step forward and evaluate LLMs' ability in higher-order ToM settings. Also,

| Component | Num. | Example |
|---|---|---|
| *Room* | 30 | kitchen 🍳, bedroom 🛏️ |
| *Object* | 37 | lemon 🍋, peach 🍑 |
| *Container* | 39 | red_envelope 🧧, blue_bottle 🍼 |
| *Agent* | 40 | Jack 🤵, Ella 👵, Noah 👨 |

Table 1: Basic components, numbers of choices for each component (Num.), and their examples in HI-TOM stories.

we are the first one pioneering in adding the deceptive communication protocol in ToM setups, which takes an initial step toward evaluating LLMs' ability in real-world scenarios. Concurrent to our work, Ma et al. (2023) surveyed the existing ToM benchmarks and conducted preliminary experiments on situated evaluation of ToM for LLMs.

## 3 The HI-TOM Dataset

To systematically examine how effectively LLMs reason Theory of Mind (ToM) at different orders, each story is coupled with five questions that require the zeroth to fourth level of ToM reasoning, respectively. Following Nematzadeh et al. (2018) and Le et al. (2019), we automatically generate HI-TOM stories. Additionally, we manually review the generated stories, questions, and answers to ensure that they are consistent with each other, and they are logically correct.

### 3.1 Dataset Design

**Story Design.** HI-TOM stories consist of four fundamental elements: rooms, objects, containers, and agents, as shown in Table 1. A story narrates events occurring in one or more rooms, where multiple objects are placed inside their respective containers. Each story features five rational agents.

Each story comprises one to three chapters. Each chapter corresponds to a single round in the object-finding game. In each chapter, we design multiple actions and optional communication protocols among agents:

- **Entry:** At least one agent enters one room, where they observe all the objects, other agents, and their actions in that room (e.g., Figure 2 Scene 1).

- **Object Movement:** When in a room, each agent can choose whether to move an object before their exit. Such actions are done in a sequential manner. In other words, the later agent can only perform such an action after the former agent

| | HI-TOM One-Chapter Story |
|---|---|
| 1 | Emma, Charlotte, Benjamin, Aiden and Isabella entered the workshop. |
| 2 | The pear is in the red_treasure_chest. |
| 3 | Emma moved the pear to the blue_suitcase. |
| 4 | Emma exited the workshop. |
| 5 | Charlotte exited the workshop. |
| 6 | Benjamin lost his watch. |
| 7 | Benjamin exited the workshop. |
| 8 | Aiden moved the pear to the blue_crate. |
| 9 | Aiden exited the workshop. |
| 10 | Isabella moved the pear to the red_treasure_chest. |
| 11 | Isabella likes the red_box. |
| 12 | Isabella exited the workshop. |
| 13 | Aiden publicly claimed that the pear is in the blue_drawer now. |
| 14 | Emma privately told Isabella that the radish is in the red_suitcase now. |

Table 2: An example HI-TOM one-chapter story with agent communications. Random distractors are inserted in lines 6 and 11, where the latter introduces "red_box" as a distractive answer choice.

moves the object (or not) and leaves the room (Scenes 2, 3, and 4).

- **Agent communication:** Outside the room, agents may be involved in two types of communications: *public*, where an agent shares information with every agent, and *private*, where an agent only speaks to another agent privately, or they can remain silent (Scene 5).

For agent communications, we set the shared information to be deceptive in order to emulate the dynamics of the complicated social life. It also adds another layer of complexity to the ToM reasoning process, which requires the answerers to not only reason about an agent's perceptions of other agents knowledge of the objects location, but also reason about whether an agent would trust another agent. In this way, we evolved the simplistic toy stories in the previous dataset (Le et al., 2019), and solved a core problem in the previous evaluations that the answers may be simply found from the objects original or final location.

Moreover, we pose a constraint that the listener would update their world knowledge based on the information given by the speaker if the speaker exits the room later than the listener. This is based on the assumption that the listener would be unaware of any changes after their exit, but the speaker might possess more up-to-date knowledge as they leave later. Additionally, we assume that Alex and Sally, who give out information publicly or privately, will believe that all the listeners trust their

respective information. A full assumption list is attached to each story, as shown in Table 7 in Appendix B.1.

Each chapter involves entry and object movement, while agent communication is optional. In Hɪ-ToM, half of the stories have at least one chapter with agent communications, while the other half only contains chapters without communications.

**Question-Answer Design.** Following Le et al. (2019), for each story, we provide *five* questions that are progressively built from lower-order questions to higher ones as shown in Table 3.

| Order | Question |
|-------|----------|
| *0th* | Where is *O* really? |
| *1st* | Where does *A1* think *O* is? |
| *2nd* | Where does *A2* think *A1* thinks *O* is? |
| *3rd* | Where does *A3* think *A2* thinks *A1* thinks *O* is? |
| *4th* | Where does *A4* think *A3* thinks *A2* thinks *A1* thinks *O* is? |

Table 3: Questions asked in a story involving object *O* and five agents. *A1* to *A4* are randomly chosen from the five agents. *0th*, *1st*, *2nd*, *3rd*, and *4th* represent the ToM orders of the questions.

Following Sap et al. (2022), we adopt the multiple-choice setting and provide the correct answer along with several distractor choices.

## 3.2 Data Generation

To generate the aforementioned ToM stories with higher-order questions among agents, we adapt the generation scripts from Nematzadeh et al. (2018), which are originally limited to first or second-order ToM stories.

Our script takes a list of story components $Rooms$, $Objects$, $Containers$, $Agents$, as well as the number of chapters $\ell$ as inputs, and outputs a story with $\ell$ chapters along with five questions from zeroth to fourth order ToM.

For the generation of each chapter, we randomly choose the story components and fit them into the chapter template. Specifically, we randomly determine whether or not each agent moves the object to another container. Then, we incorporate agent communications in certain chapters, where we use the phrases "publicly claim" and "privately tell" to encode public and private communications.

To generate the questions and answers for each story, we integrate the relevant story components into a predefined question template. Subsequently, we utilize an answer generator to track the actions

| Datasets | *ToM/ToM-easy* | ToMI | Hɪ-ToM |
|----------|----------------|------|--------|
| 1st | ✓ | ✓ | ✓ |
| 2nd | ✓ | ✓ | ✓ |
| 3rd | ✗ | ✗ | ✓ |
| 4th | ✗ | ✗ | ✓ |
| Comm. | ✗ | ✗ | ✓ |
| #Line | 15.05 | 8.86 | 26.47 |
| #Agent | 3.22 | 2.75 | 5 |
| #Container | 5 | 2 | 7.39 |

Table 4: Comparison between Hɪ-ToM and other datasets. 1st, 2nd, 3rd, and 4th refer to whether a dataset contains story-question pairs of a specific ToM order. Comm. stands for the existence of agent communications. #Line, #Agent, and #Container represent average number per story.

of all agents and derive the correct answer to each question. Further details and pseudocode related to our story generation process can be found in Appendix B.2.

Additionally, based on Le et al. (2019), we further incorporate distractor sentences that relate an agent with a random container, such as "Jack likes the red_container". This reduces the regularity and predictability of the stories. Table 2 shows an example one-chapter story with agent communications and random distractors.

## 3.3 Dataset Characteristics

Table 4 shows a comparison between Hɪ-ToM and the other ToM datasets. First, unlike previous datasets, Hɪ-ToM is the only benchmark that contains third and fourth-order stories, which suggests that Hɪ-ToM is more challenging and requires higher-order ToM reasoning. Also, we first introduce communications among agents, which poses greater challenges to LLMs to reason about human interactions. In addition, stories in Hɪ-ToM are significantly longer, with a larger number of agents and containers per story. This requires the LLMs' capability to comprehend the complete storyline and reason about each agent's beliefs.

Notably, Hɪ-ToM features a larger pool of potential answers and a balanced distribution of correct answers throughout the story. In *ToM/ToM-easy* and ToMI, all the correct answers appear within the last two containers or the only two containers in the corresponding story. In contrast, the proportions of correct answers appearing in the first, second, third, and final quarters of the Hɪ-ToM stories are 28.7%, 27.2%, 18.8%, and 25.3%, respectively. The even distribution of correct answers

eliminates the position bias of correct answers being concentrated in specific segments of the stories in HI-TOM.

## 4 Experimental Setup

### 4.1 Models

We evaluate the following four LLMs on HI-TOM:

1. **GPT-3.5-Turbo** (OpenAI, 2022) and **GPT-4** (OpenAI, 2023) are closed-sourced models from OpenAI. We use `gpt-4-32k` and `gpt-3.5-turbo` for experiments, which are conducted on June 14th∼15th 2023.
2. **Claude-instant** is a close-sourced model published by Anthropic[†].
3. **Guanaco** (65B) is an open-sourced model fine-tuned from LLaMA (Touvron et al., 2023).

We adhere to the default parameter configurations across all the examined language models.

### 4.2 Methods

For each HI-TOM story, we conduct trials using two prompting styles: *Vanilla Prompting* (VP) and *Chain-of-Thought Prompting* (CoTP). In VP prompting, the model needs to pick the best answer from a given set of options without explanation. CoTP prompting requires the model to offer a step-by-step explanation of its thought process along with the answer. Appendix C.1 provides an example (Table 8) and more details of our prompting methods.

### 4.3 Evaluation

We evaluate the model performance using both *standard accuracy* (hereafter referred to as *accuracy*) and *joint accuracy*. Adapted from (Le et al., 2019), *joint accuracy* represents a more stringent metric than *standard accuracy*. It considers an answer as correct only when the related question, along with all preceding, lower-order questions within the same story are answered correctly. For instance, the third-order question in Table 3 is considered correct only if the model correctly answers the zeroth, first, second, and third-order questions above it. *Joint accuracy* effectively reveals the model's genuine ability in higher-order ToM reasoning, as the model may only reason the higher-order ToM correctly if it is able to reason the lower-order ToM because the higher-order question is a

---
[†]www.anthropic.com/index/introducing-claude

| Model & Methods | | Accuracy (%) | | | |
|---|---|---|---|---|---|
| | | w/o dec. | w/ dec. | Overall | |
| Guanaco 65B | VP | 33.33 | 33.33 | 33.33 | 32.17 |
| | CoTP | 35.00 | 26.99 | 30.99 | |
| | | +1.67 | -6.34 | -2.34 | |
| Claude -instant | VP | 49.33 | 42.00 | 45.67 | 46.00 |
| | CoTP | 52.33 | 40.33 | 46.33 | |
| | | +3.00 | -1.67 | +0.66 | |
| GPT-3.5 -turbo | VP | 28.67 | 26.33 | 27.50 | 31.50 |
| | CoTP | 35.67 | 35.33 | 35.50 | |
| | | +7.00 | +9.00 | +8.00 | |
| GPT-4 -32k | VP | 60.42 | 55.81 | 58.11 | 58.99 |
| | CoTP | 64.04 | 55.72 | 59.88 | |
| | | +3.60 | -0.09 | +1.77 | |

Table 5: Standard accuracy results of the four tested models on HI-TOM stories. "w/o dec." and "w/ dec." indicate accuracy in stories with and without deception, respectively. The performance increase and decrease from VP to CoTP prompting style are highlighted.

recursive successor of the lower-order ones for the same story.

## 5 Experimental Results

Table 5 presents the accuracy scores of the four LLMs. All the LLMs we evaluate exhibit less than 60% accuracy scores, demonstrating that HI-TOM is challenging even for the most sophisticated LLMs. Figure 3 depicts the joint accuracy scores of GPT-4 and GPT-3.5 under various settings. As the story length decreases or the ToM order increases, LLMs' performance decreases across various settings. In addition, LLMs perform worse when there are deceptive agent communications involved in the story. The trend observed in Guanaco and Claude aligns with that of GPT-4 and GPT-3.5, as shown in Appendix C.2.

The experimental results also reveal the following noteworthy patterns:

**CoTP prompting yields insignificant performance gains.** We observe no substantial improvement in accuracy when transitioning from VP to CoTP in 5. Furthermore, in the assessments involving stories with deception, the switch in prompting methods even leads to a decrease in accuracy. We hypothesize that as there are more steps involved, there are higher chances of deceptive information misleading steps in between. The chain may then amplify the error in that step, leading to a cascade of errors throughout the reasoning process.

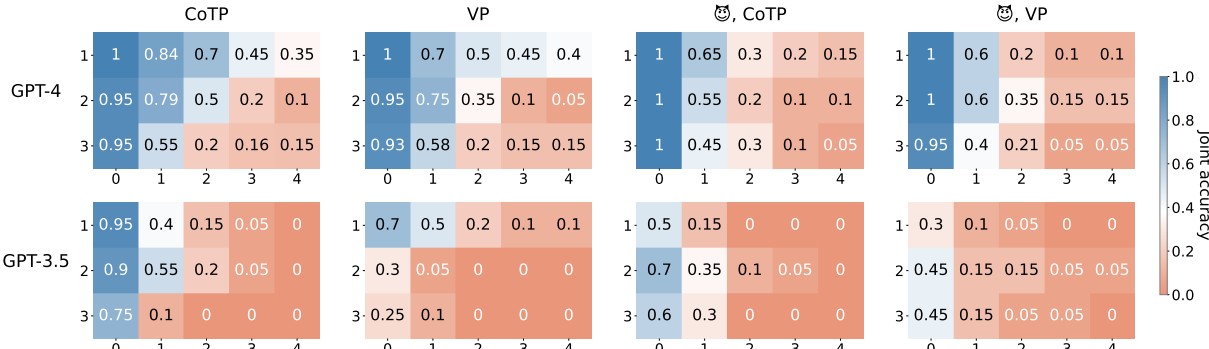

Figure 3: Joint accuracy of GPT-4 and GPT-3.5 on HI-TOM stories w/ or w/o deceptive agent communications. The $x$-axis stands for ToM orders, and the $y$-axis is for story lengths (number of chapters). CoTP and VP respectively represent chain-of-thought and multiple-choice-w/o-explanation prompting styles. The devil sign (😈) signifies accuracy results on stories with deception, while other results pertain to non-deceptive stories.

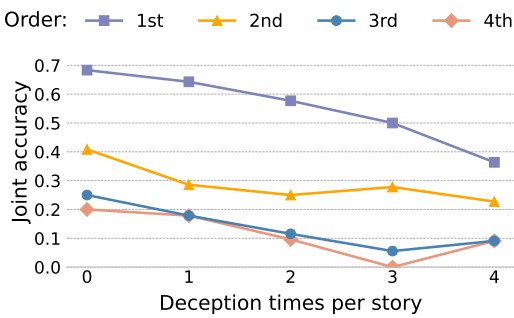

Figure 4: Joint accuracy of GPT-4 on HI-TOM stories with 0 to 4 sentences of deceptive agent communication. 0th-order (reality) accuracy is not included, since the answer to the real room of the objects is not affected by deceptive communications.

**Increased ToM order leads to decreased performances.** As the ToM order increases from the zeroth to the fourth, the joint accuracy goes from near perfect to near zero. We also observe the drastic decline in the conventional accuracy scores in Appendix C.2.

**LLMs' performance decreases as there are more deception communications involved.** The performance drops when deception communications are involved. Table 5 and Figure 3 reveals a worse performance on stories with agent communications. To further investigate the models' reasoning abilities in handling deceptive agent communications, we plot the resulting accuracy versus the number of deceptive communication sentences ("deception times") per story for GPT-4 in Figure 4. As shown in Figure 4, as deception times increase from 0 to 4, the joint accuracy experiences drops of 32%, 18.1%, 16%, and 11% respectively for the four ToM orders. This suggests that the deceptive agent

communications challenge the LLMs in their ToM reasoning process.

# 6 Discussion and Analyses

## 6.1 Underlying Patterns in Correct LLM Predictions

Although the overall accuracy of the models leaves room for improvement, we observe a higher frequency of correct model choices under specific conditions. We thus examine the scenarios where models have higher answer accuracy.

**LLMs handle answers that appear at the beginning and end better.** When dealing with long three-chapter stories, LLMs frequently overlook key information, such as the movement of a specific container or agent conversations. Yet, they tend to pay special attention to the beginning and the end of the story.

In Figure 5, we highlight GPT-4's higher performance when the correct answer aligns with the first or last container mentioned in the story, as compared to other cases, as demonstrated by the higher values on the diagonal. This suggests that LLMs are better at handling answers that appear at the beginning or at the end. In contrast, The accuracy when the correct answers are the middle containers (i.e. neither the first nor the last) is similar to those that are not in those containers, as shown in Figure 14 in Appendix C.2. We observe similar patterns of the Claude model focusing on the beginning and end of stories, as shown in Figure 15 in Appendix C.2. Our findings about position bias in LLMs align with other works on LLMs (Wang et al., 2023).

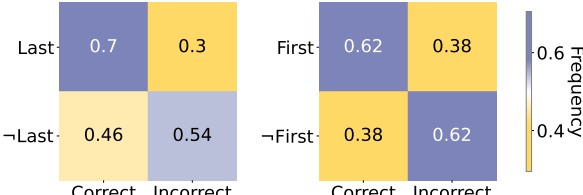

Figure 5: Frequency of GPT-4 correctly or incorrectly answering a question of a three-chapter story, based on whether or not the correct answer is the last or first container mentioned in the story. "Last"/"First" and "¬Last"/"¬First" indicate whether or not the correct answer lies at the last/first container.

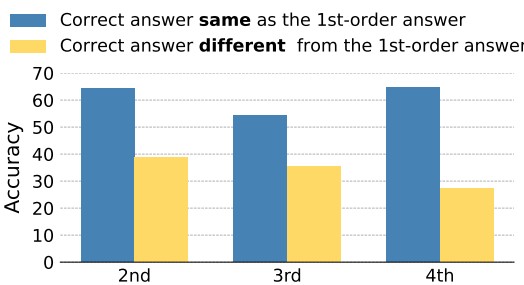

Figure 6: Standard accuracy of GPT-4 on 2nd, 3rd, and 4th-order questions, categorized by whether the correct answer matches the corresponding 1st-order answer.

**LLMs perform better if the answers across orders are the same.** We observe that LLMs perform better on question sets where the higher-order answer coincides with a lower-order answer. In Figure 6, we see a clear performance disparity between LLMs answering correctly if the answers are the same across orders versus the answers being different across orders. However, this may result from LLMs' tendency to predict the same answers across orders. We find that 72.4%, 64.6%, and 59.8% of GPT-4's second, third, and fourth-order answers match their first-order responses. In contrast, only 30.9%, 20.9%, and 22.2% of the corresponding correct answers in HI-TOM are the same as their first-order answers. This suggests that GPT-4's enhanced performance on certain questions may be due to the coincidence of correct answers across different ToM orders.

### 6.2 Classifying Reasoning Errors

To provide a comprehensive overview of the failure cases of LLMs in ToM reasoning, we manually evaluate a total of 300 step-by-step responses across all ToM orders by ourselves, comprising 150 from each of GPT-4 and GPT-3.5. Table 6 describes the five most prevalent error types with corresponding examples. Figure 7 provides the frequencies of these errors in GPT-4's responses across different orders. We also show the results for GPT-3.5 and do a comparison between the two LLMs in Appendix C.2. As the ToM order increases, LLMs tend to demonstrate a higher frequency of errors. Here we provide hypotheses and discussions for each of the error types:

**Insufficient reasoning depth.** We notice that LLMs tend to skip steps in their reasoning process and end up with an answer to a lower-order question, as we observed earlier in Section 6.1. One reason can be that the pre-training corpus often

consists of simple patterns rather than complex and nuanced reasoning scenarios, leading to its frequent simplification of the questions. In addition, LLMs may possess a limited contextual understanding of the story. They may struggle to retain and integrate information from multiple steps or make connections across different parts of the text, leading to oversimplification of the question.

**Commonsense errors.** LLMs have demonstrated remarkable performance on standard benchmarks of commonsense reasoning (Bian et al., 2023). However, when it comes to ToM reasoning, even advanced models like GPT-4 are prone to making mistakes in handling commonsense knowledge. One key aspect that contributes to these errors is the disparity between the models' knowledge of commonsense facts and their ability to effectively apply that knowledge in the complex reasoning process. While LLMs may possess a vast amount of explicit commonsense knowledge, they can struggle to appropriately utilize this knowledge while avoiding overgeneralization. In addition, the frequent commonsense errors in HI-TOM might be due to that HI-TOM is newly constructed, and therefore LLMs have never seen such data before. In contrast, LLMs' pre-training corpus might contain the data in the publicly available commonsense benchmarks, leading to the high performances on LLMs on those benchmarks (Magar and Schwartz, 2022).

**Hallucinations.** Hallucination is a well-known phenomenon in LLMs' generation process (McKenna et al., 2023). In our experiments, LLMs may have relied on superficial cues and statistical associations to answer the questions, rather than gaining a solid understanding of the underlying context and meaning. Hence, they might resort to fabricating baseless details to bridge the logic gap between the true story and their erroneous responses.

| Error Types | Description | Example |
|---|---|---|
| ▶ *Insufficient Reasoning-Depth* | Oversimplify the question and skip the required multi-step reasoning. | ⍰ : Where does Jack think Hannah thinks William thinks the carrot is? 
 🤖 : green_box, as that's `where Jack last saw it.` |
| ▶ *Commonsense Errors* | Generate outputs that violate common sense. | 🤖 : . . . Aiden exited the pantry after step 8, but he can `still witness` the move in the pantry `after the exit`, so . . . |
| ▶ *Hallucinations* | Fabricate ungrounded details or facts. | (In the story, Benjamin saw a cat, but `did not talk` about it) 
 🤖 : . . . But there's another twist. Suppose Ella also learns that Benjamin `lied about seeing a cat` to distract everyone from his real plan . . . |
| ▶ *Temporal Ignorance* | Confuse or ignore the temporal order of events. | 🤖 : . . . Lily exited the hallway (step 8) `after` Amelia moved the corn to the red_basket (step 11), . . . |
| ▶ *Spurious Causal Inference* | Attribute a cause-and-effect relationship between unrelated events. | 🤖 : . . . Carter privately told Emma that the tomato is in the green_drawer. Private communications are not heard by others, `so` Emma has no reason to doubt Carter's information. |

Table 6: Types of reasoning errors commonly made by LLMs, with their description and example erroneous responses (🤖) to questions (⍰) from our experiment results on GPT-4.

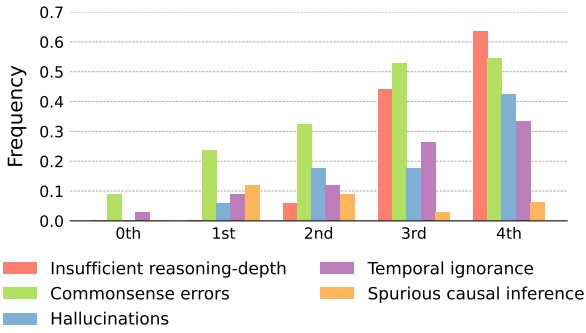

Figure 7: Ratio of GPT-4 answers containing the five reasoning errors. The $x$-axis corresponds to ToM orders.

**Lack of temporal information.** We observe that LLMs' understanding of the sequence of agent actions is often skewed, as the actions are closely listed in HI-TOM stories. This confusion in temporal order is also found in Yuan et al. (2023). This error may be attributed to biases inherent in the pre-training corpus which leads to LLMs' lack of genuine understanding of temporal relations.

**Spurious Causal Inference.** The current learning paradigm for LLMs is designed to capture the statistical correlations among the data (Devlin et al., 2019). Through such a paradigm, it is difficult for LLMs to capture the underlying logic behind these correlations (Jin et al., 2023). As a result, LLMs may make incorrect or misleading causal inferences based solely on these superficial patterns.

## 7 Implications on the Future of NLP

We believe our work has important implications on the future directions of NLP with respect to the two-way relation between artificial intelligence and human intelligence.

**Enhance LLMs' ToM ability from the perspective of human intelligence.** According to Kahneman (2011), human decisions are supported and guided by the cooperation of two capabilities or two systems: System 1 for intuitive, imprecise, fast, and often unconscious decisions ("thinking fast"), and System 2 for more complex situations with logical and rational thinking ("thinking slow"). This theory has inspired works in computer vision and natural language processing communities to explicitly equip models with the two systems (Hill et al., 2021; Miech et al., 2021).

Through our examination of LLMs' ToM ability, we find failure cases that resemble the characteristics of System 1 thinking; for instance, LLMs may invent causes and intentions ("hallucination"), or substitute an easier question for a difficult one ("insufficient reasoning-depth"). Furthermore, the significant performance drop from zeroth to fourth order ToM in HI-TOM suggests that LLMs may be more inclined to System 1 thinking rather than System 2 thinking, as higher-order ToM requires careful in-depth logical inference.

However, there exists a line of research combining the symbolic reasoning process (Simon and Newell, 1971; Winograd, 1971), which aligns with System 2 thinking, with connectionist paradigm or neural learning (Rumelhart et al., 1986; LeCun et al., 2015), which captures the intuitive and pattern recognition aspects of System 1 thinking

(Shavlik, 1994; Hitzler, 2022). This integration holds the promise of enabling AI systems to perform complex tasks that require both logical deduction and statistical generalization. We believe our findings of ToM limitations of LLMs alongside this previous line of research clearly points to a direction where neural and symbolic approaches are combined in order to achieve abilities that are more closely aligned to human intelligence.

**Understanding humans through the lens of LLMs.** ToM plays a crucial role in understanding human intelligence, as it is an important aspect of human cognition that enables us to make inferences about others' thoughts, emotions, and behaviors. Enabling progress in ToM reasoning in LLMs entails progress in emulating the functioning of human mind, which in turn offers intriguing possibilities for gaining insights into human interactions and the emergence of intelligence. While it is important to recognize that the analogy between human and artificial intelligence has its limitations and is a subject of debate within the NLP community (Bender et al., 2021), recent research has explored the extrinsic understanding of human interactions through multi-agent systems (Park et al., 2023). This approach allows us to observe how LLMs can mimic and simulate aspects of human behavior and communication. By studying LLMs and their intrinsic properties, such as the emergence of intelligence, we can gain valuable insights into the fundamental processes underlying human cognition (Wijmans et al., 2023). Researchers have also developed methods to elicit human-like behavior from LLMs, providing further opportunities to explore and understand the capabilities and limitations of these models (Belrose et al., 2023). While LLMs offer a close-up view of human-like language processing, it is crucial to approach the topic with caution and recognize the complexities and nuances of human intelligence and behavior.

**Enhance LLMs' ToM abilities for better NLP applications.** In daily life, our ToM ability plays a vital role in understanding others' intentions, therefore helping us in our communication. In HI-TOM, we enable higher-order ToM reasoning, which in turn can lead to improvements in LLMs performance on tasks such as deception detection, emotional support, empathetic communication, and others. Additionally, since LLMs represent foundational models that are used across various NLP tasks and applications, enhancing the abilities of LLMs opens up exciting possibilities for improving specific NLP tasks that benefit from these models.

## 8 Conclusion

In this paper, we introduce HI-TOM, the first ToM benchmark that contains higher-order ToM tasks. We demonstrated that LLMs' performance suffers a significant drop in ToM tasks from lower to higher order. By proposing HI-TOM, we hope to address the challenges of ToM in complicated scenarios and spark further research on enhancing the reasoning ability of LLMs.

Furthermore, we present our insights on the future of NLP and discuss potential directions for enhancing LLMs. Our aim is to stimulate research that draws inspiration from human intelligence, strives to understand humans better, and ultimately leads to the development of NLP applications that better cater to the needs of humans.

## Acknowledgement

We thank the anonymous reviewers for their valuable feedback and discussion. This paper's draft version was accepted to the non-archival track of the ToM workshop at ICML 2023. We would also like to extend our appreciation to the reviewers from the ToM workshop for their feedback.

## Ethical Considerations

The data used in our study were collected from the API of language models. No sensitive or personal information was included.

Our research aims at examining the high-order ToM reasoning ability of LLMs, and we demonstrate the insufficient higher-order ToM abilities of the current LLMs. There is no direct misuse of our findings. However, we recognize that future LLMs with stronger ToM abilities may become more powerful at generating misinformation and even manipulating people if used by some ill-intended parties. Therefore, we advocate for the responsible use of LLMs and the associated technologies.

We adhere to the principles of transparency and openness. All methods and findings are reported completely and honestly. Furthermore, we will make our code public upon acceptance. We invite readers to utilize this resource for a more comprehensive understanding of our methods and results.

## Limitations

The limitations of our work can be stated from the following perspectives.

1. Due to the constraint of computing resources and budget, we only test four LLMs. However, we try our best to select the representative LLMs from close-sourced to open-sourced LLMs including GPT-3.5 and GPT-4 from OpenAI, Claude from Anthropic, and Guanaco from the community.

2. Due to the scope of this paper, we only demonstrate the insufficient ToM abilities of LLMs. Future works may further investigate how different training paradigms such as training with or without reinforcement learning with human feedback (RLHF) affect the ToM ability of these LLMs.

3. We acknowledge that our dataset was constructed based on specific rules, which means its dialog syntax may differ from genuine conversations. In the real world, higher-order interactions might occur in a more implicit manner, embedded within more intricate dialogues and questions. We plan to address this in future research.

4. We share our thoughts on the future of NLP and research with LLMs, hoping to stimulate research that draws inspiration from human intelligence, understands humans better, and serves humans better. We admit that there exist alternative ways of moving forward on NLP research. We welcome feedback and open discussion on how we can collectively advance NLP research.

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

## A  Author Contributions

Yufan Wu and Yinghui He conceived of the idea and planned the experiments. Yufan Wu and Yinghui He took the lead in writing the script for dataset generation. Yilin Jia, Yufan Wu, and Yinghui He carried out the experiments of testing language models. Yulong Chen and Naihao Deng supervised the project. Naihao Deng came up with the high-level idea for the paper, which was later refined in team discussions. All authors contributed to the writing and editing of this paper. Specifically, Yufan Wu and Yinghui He drafted the paper. Yilin Jia contributed to the writing for experiment setups. Rada Mihalcea, Yulong Chen, and Naihao Deng helped edit the paper.

## B  HI-TOM Details

### B.1  Assumptions

Our simplified deception and belief mechanisms are based on four assumptions. Table 7 shows the original assumption list we attach to each story in the dataset and prompt into LLMs.

```
Note: You should assume the following.
(1) An agent witnesses everything and every
movement before exiting a room.
(2) An agent A can infer another agent B's
mental state only if A and B have been in
the same room, or have private or public
interactions.
(3) Note that every agent tend to lie. What
a character tells others doesn't affect his
actual belief. An agent tend to trust a agent
that exited the room later than himself. The
exit order is known to all agents.
(4) Agents in private communications know that
others won't hear them, but they know that
anyone can hear any public claims.
```

Table 7: Assumption list attached to each HI-TOM story and prompt into LLMs.

### B.2  Story Generation Details

Algorithm 1 and Algorithm 2 provide the pseudocode for the generation process of each chapter and the whole story in HI-TOM.

In Algorithm 1, the function MOVE is employed to populate the story components into the template, thereby producing a sentence that describes the movement. The function COMMUNICATE generates content related to the "tell" action. Meanwhile, the RANDOM_DISTRACTOR function introduces random distractors into the story.

In Algorithm 2, the question generator Q_GEN randomly picks the agents and the object appearing in the story and populates them into a predefined question template. Then, the answer generator A_GEN generates the answer to the corresponding question based on a dictionary that traces the beliefs of different orders of each agent.

---

**Algorithm 1** HI-TOM Chapter Generation Algorithm

**Input:** $agents, room, conts, obj$
**Output:** $chap$
1: **function** CHAP($agents, room, conts, obj$)
2:     **for** $agent$ in $agents$ **do**
3:         set $no\_move$ to random boolean value
4:         $move \leftarrow$ MOVE($agent, conts, obj, no\_move$)
5:         add $move$ into $chap$
6:     **end for**
7:     $com \leftarrow$ COMMUNICATE($agents$)
8:     $rd \leftarrow$ RANDOM_DISTRACTOR($agents, conts$)
9:     add $com$ and $rd$ into $chap$
10:     **return** $chap$
11: **end function**

---

## C   Experiment Details

### C.1   Prompting inputs

In our experiments, the average number of tokens in a single prompt is 453.3, and the total token number of our prompts on each model is 543968, including VP and CoTP prompts on stories with or without deception.

Table 8 is a sample CoTP prompt in our experiments. We specify the range of the story, question, choices, and assumptions to enhance the models' understanding. We also order each line of the story to indicate the chronological order. The provided answer choices are all the containers appearing in the story.

### C.2   Supplementary Results

Figure 9 and Figure 8 shows the detailed joint accuracy results of Guanaco and Claude. The joint accuracy generally decreases as the story length and the ToM orders increase, aligning with the results of GPT-4 and GPT-3.5. The overall joint accuracy performance of Claude is better than Guanaco.

Figure 10 to Figure 13 show the standard accuracy results of GPT-4, GPT-3.5-turbo, Claude-instant and Guanaco 65B, as the break-down details of Table 5. Among the four models, GPT-4 has the highest and most stable performance, reaching nearly perfect accuracy on the zeroth order and higher than 20% on the fourth.

Under CoTP prompting, each model reaches a high performance on zeroth-order questions, espe-

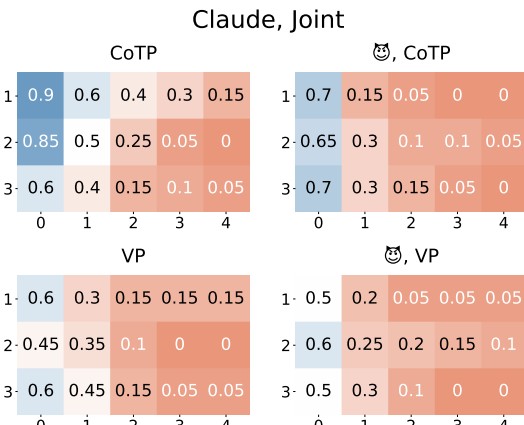

Figure 8: Joint accuracy results of Claude-instant.

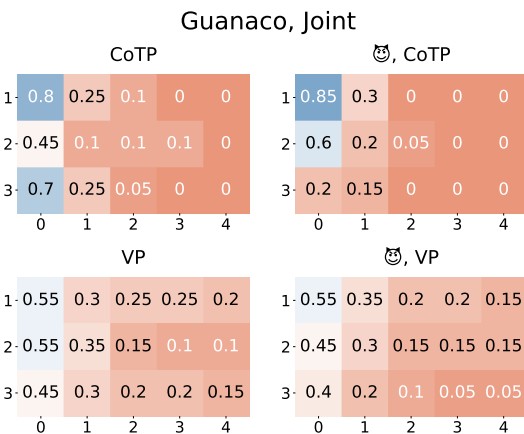

Figure 9: Joint accuracy results of Guanaco 65B.

```
Read the following story and answer the multiple-choice question.  Think step-by-step.
Provide the answer first, and then explain it.
Story:
1 William, Jack, Charlotte, Noah and Hannah entered the hall.
2 Noah saw a monkey.
3 The carrot is in the red_basket.
4 William made no movements and stayed in the hall for 1 minute.
5 William exited the hall.
6 Hannah saw a mouse.
7 Jack made no movements and stayed in the hall for 1 minute.
8 Jack exited the hall.
9 Charlotte made no movements and stayed in the hall for 1 minute.
10 Charlotte exited the hall.
11 Charlotte dislikes the tomato.
12 Noah moved the carrot to the green_envelope.
13 Noah exited the hall.
14 Hannah moved the carrot to the red_basket.
15 Hannah exited the hall.
16 William, Jack, Charlotte, Noah and Hannah entered the waiting_room.
17 Charlotte publicly claimed that carrot is in the green_envelope.
18 Hannah privately told Charlotte that the carrot is in the blue_container.
Question: Where does Charlotte think Jack thinks Hannah thinks William thinks the carrot
is?
Choices: A. green_envelope, B. red_basket, C. blue_container, D. red_crate, E. green_drawer,
F. blue_bucket, G. green_cupboard, H. red_bottle, I. green_treasure_chest, J. blue_cupboard,
K. red_pantry, L. red_container, M. blue_bathtub, N. red_envelope, O. blue_pantry

Note:  You should assume the following.  (1) An agent witnesses everything and every
movements before exiting a room. (2) An agent A can infer another agent B's mental state
only if A and B have been in the same room, or have private or public interactions. (3)
Note that every agent tend to lie. What a character tells others doesn't affect his actual
belief. An agent tend to trust a agent that exited the room later than himself. The exit
order is known to all agents. (4) Agents in private communications know that others won't
hear them, but they know that anyone can hear any public claims.
```

Table 8: An example CoTP prompt of a one-chapter HI-TOM story with a fourth-order question.

---

**Algorithm 2** HI-TOM Story Generation Algorithm

**Input:** Number of chapters: $\ell \in \{1, 2, 3\}$
         Story components: $Rooms, Objects,$
                 $Containers, Agents$
**Output:** $story, question, answer$
1: **function** STORY($\ell, Rooms, Objects, Containers,$
   $Agents$)
2:    **for** $i \leftarrow 1$ to $l$ **do**
3:        randomly choose $room, obj, conts, agents$
4:        $chap \leftarrow$ CHAP($room, obj, conts, agents$)
5:        add $chap$ into $story$
6:    **end for**
7:    $question \leftarrow$ Q_GEN($Agents, Objects$)
8:    $answer \leftarrow$ A_GEN($Agents, Objects$)
9:    **return** $story, question, answer$
10: **end function**

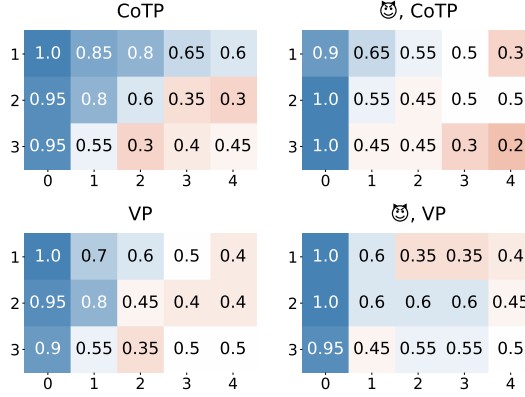

Figure 10: Standard accuracy results of GPT-4.

cially for stories without agent communications, and their performance deteriorates with increased ToM order and story length. Yet, under VP prompting, Claude and Guanaco exhibit a uniform performance of around 50% across all the orders.

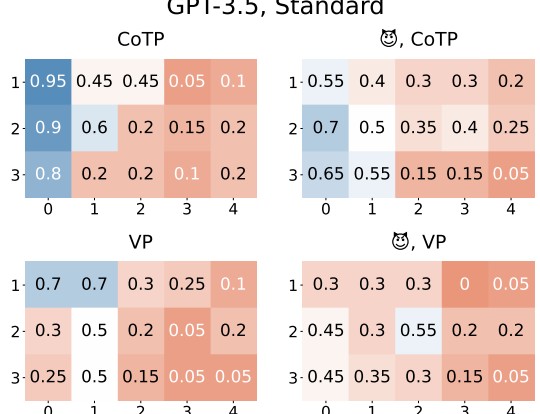

Figure 11: Standard accuracy results of GPT-3.5-turbo.

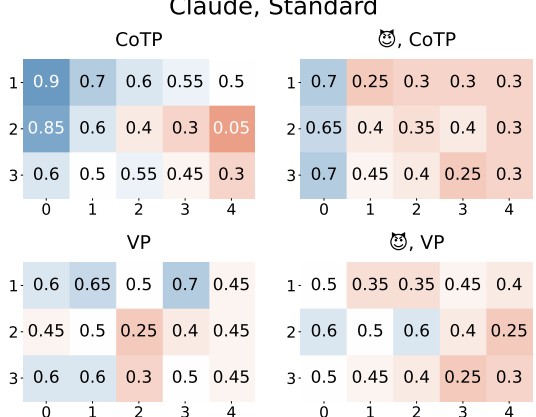

Figure 12: Standard accuracy results of Claude-instant.

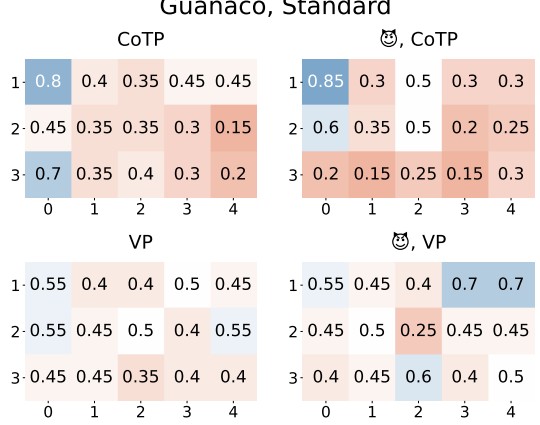

Figure 13: Standard accuracy results of Guanaco 65B.

Figure 14 illustrates the performance comparison of GPT-4 between the case when the correct answer is in a certain position in the middle of the story, and the case when it is not. We observe that GPT-4 does not significantly perform better when the correct answer lies in the middle of the story. This serves as a contrast to Figure 5, highlighting the better ability of GPT-4 to capture answers at the beginning or the end of a story.

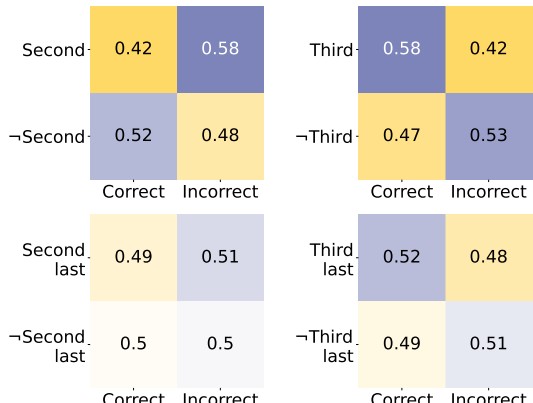

Figure 14: Performance of GPT-4 in the cases when the correct answer does or does not lie in a certain position.

Figure 15 shows similar observations for Claude. The plots for the last and first positions of containers show a higher frequency in the top-left and bottom-right cells, while the plots for other positions do not imply such a pattern.

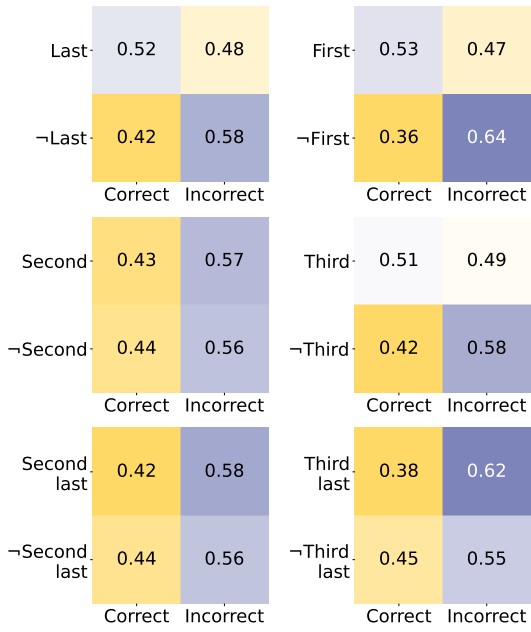

Figure 15: Performance of Claude in the cases when the correct answer does or does not lie in a certain position.

Figure 16 details the appearance frequency of the five reasoning errors in the step-by-step responses of GPT-3.5. Compared to the error ratios of GPT-4 (Figure 7), the frequencies of commonsense errors, hallucinations, and spurious causal inference are significantly higher, implying GPT-3.5's immature perceptions of the world and its deficient logical reasoning abilities. The occurrence of insufficient reasoning depth and temporal ignorance escalates

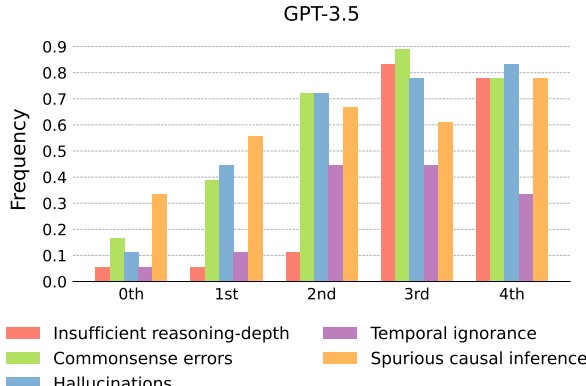

Figure 16: Ratio of the five reasoning errors in GPT-3.5's responses.

in higher-order responses.

The comparison between Figure 7 and Figure 16 yields that GPT-4 has not resolved the errors of commonsense, insufficient reasoning depth, and temporal ignorance, while hallucinations and spurious causal inference have been largely addressed.