# OpenReview forum: "Hi-ToM: A Benchmark for Evaluating Higher-Order Theory of Mind Reasoning in Large Language Models"
_EMNLP/2023/Conference — EMNLP 2023 Findings_

### Official Review · Reviewer_ZmFn · 2023-08-02

**Soundness:** 3

**Excitement:**

3: Ambivalent: It has merits (e.g., it reports state-of-the-art results, the idea is nice), but there are key weaknesses (e.g., it describes incremental work), and it can significantly benefit from another round of revision. However, I won't object to accepting it if my co-reviewers champion it.

**Paper Topic And Main Contributions:**

The paper introduces Hi-ToM, a new benchmark for evaluating higher-order Theory of Mind (ToM) reasoning in large language models (LLMs). It contains stories with questions up to fourth-order ToM, incorporating agent communications. The stories are longer with more agents and containers compared to prior ToM datasets, requiring comprehending the complete storyline. The dataset is generated using scripts adapted from prior work, with manual verification for quality. Experiments using GPT-3.5, GPT-4, Claude, and Guanaco show performance declines as ToM order increases, especially with deceptive communications, demonstrating limitations in recursive reasoning. Failure analysis reveals insufficient reasoning depth, temporal ignorance, and other deficiencies. The key contribution is the new higher-order ToM benchmark that highlights challenges for LLMs in complex multi-agent reasoning.

**Reasons To Accept:**

1. The authors explore a higher-order Theory of Mind (ToM) benchmark by constructing and validating ToM stories along with corresponding questions and answers. This has the potential to become a valued resource for evaluating the cognitive abilities of LLMs.
2. The authors evaluate cutting-edge LLMs including GPT-4, GPT-3.5-Turbo, Claude, and Guanaco. They demonstrate that the performance of LLMs on the proposed higher-order ToM benchmark is far from perfect.
3. The authors provide detailed analysis of the experimental results and offer insights into how ToM-related research could influence future developments in the LLM field. Their discussion is thoughtful and insightful.

**Reasons To Reject:**

1. A major concern is whether such story-grounded question-answering benchmarks can genuinely assess LLMs' ToM abilities. As discussed in the paper "Clever Hans or Neural Theory of Mind? Stress Testing Social Reasoning in Large Language Models", LLMs tend to answer questions based on linguistic patterns and spurious correlations rather than actual cognitive processes. The authors should address this issue with proper discussion.
2. Based on Table 1, the proposed Hi-Tom dataset is relatively small, containing only 146 samples. This raises concerns about the validity of evaluation. Additionally, the authors only present pseudocode for the question and answer generators without explaining their actual implementation details.
3. In designing agent communications, the authors introduce two modes: public and private. They set that the shared information is deceptive. However, the authors do not sufficiently explain the motivation behind incorporating deceptive information. While LLMs' performance decreases with more deceptive communications, this seems obvious and does not justify the necessity of deception.

**Reproducibility:**

3: Could reproduce the results with some difficulty. The settings of parameters are underspecified or subjectively determined; the training/evaluation data are not widely available.

**Reviewer Confidence:**

4: Quite sure. I tried to check the important points carefully. It's unlikely, though conceivable, that I missed something that should affect my ratings.

---

> ### Author Rebuttal · Authors · 2023-08-29
>
> Thank you for your valuable review. We are glad you found our work interesting and the discussion thoughtful and insightful. Below, we address each of your concerns:
>
>
> - **Response on whether we’re assessing LMs’ genuine ToM ability (weakness 1):**
> 	-   We concur that current LLMs tend to capture spurious patterns from the context as well as the question. Therefore, we made some efforts on the data construction to mitigate the impact of spurious patterns:
> 		-   We made sure that the correct answers to the questions were not only located at the beginning or the end of the stories, but were evenly distributed (as mentioned in Section 2.3, Line 214-221). This prevents LLMs from hitting high accuracy by consistently choosing the original or final location of the object.
> 		-   We informed LLMs of the common sense and assumptions used in our story setting at the end of each query (Appendix B.1, Line 800-804). This discourages LLMs from generating unreasonable stories by overlooking obvious facts (Shapira et al., 2023), and avoids obtaining a very low accuracy.
> 	- While LLMs indeed have a tendency to exploit data patterns, as mentioned in Shapira et al. (2023), our benchmark aims to present a more intricate challenge that is harder to solve through mere pattern recognition. We believe the unpredictable nature of our stories and questions helps reflect a more genuine form of ToM reasoning.
>
>
>
> -   **Response on lack of samples (weakness 2):**
> 	-   We would like to point out that the size of our dataset (number of stories) is **1.2k**, instead of 146. The numbers in Table 1 reflect the numbers of different story components. For example, “30 rooms” means that there are 30 possible choices of room names (e.g. Kitchen) in our stories.
> 	-   The 1.2k size is comparable to the size of the test set in the previous dataset (ToMi, 2023), so we believe that the dataset is large enough for a valid evaluation. We will highlight these clarifications in our revision.
>
>
>
>
> -   **Response on lack of implementation details for the question and answer generator (weakness 2):**
> 	-   We agree that we need to explain more clearly how we implemented the question and answer generators, and we will modify Line 816-820 in the camera-ready version as follows:
> 		-   For the generation of a question, we randomly select an object and several names, then populate them into the pre-defined question template.
> 		-   To produce the answer, we construct a comprehensive dictionary that documents the position of each object and the multi-layer beliefs of every agent. The answer is then directly retrieved using the names and objects selected into the question.
> 	-   For the detailed code, our implementation of the question and answer generator inherits the publicly released code of the paper “Evaluating Theory of Mind in Question Answering” (2018), as we specified in the Data Generation Section (Line 163-164). Also, we plan to release our code upon acceptance, as mentioned in our paper (Line 555-556).
>
>
>
>
> -   **Response on the motivation of deception (weakness 3):**
> We understand that a motivation for the deceptive communication should be presented, and we will add additional explanations for the motivation in the revised version as follows:
> 	-  Our story setting was created to emulate the dynamics of the complicated social life, where sophisticated people take different actions (agents move the object) and strategically communicate with each other (agents conduct deceptive communications).
> 	- Also, deceptive communications can add another layer of complexity to the ToM reasoning process, which requires LLMs to not only reason about agents’ perceptions of other agents’’ knowledge of the object’s location, but also reason about whether an agent would trust another agent. In this way, we evolved the simplistic toy stories in the previous dataset (ToMi, 2019), and solved a core problem in the previous evaluations that the answers may be simply found from the object’s original or final location.
>
> -   **Response on reproducibility:**
> We will make our datasets as well as the generation code publicly available upon acceptance.
>
>
>
> We hope we clarified your concerns, and we would be happy to engage in further discussions as needed.
>
> **Reference**
>
> Shapira, Natalie, et al. "Clever hans or neural theory of mind? stress testing social reasoning in large language models." arXiv preprint arXiv:2305.14763 (2023).

---

### Official Review · Reviewer_Trb7 · 2023-08-04

**Soundness:** 4

**Excitement:**

3: Ambivalent: It has merits (e.g., it reports state-of-the-art results, the idea is nice), but there are key weaknesses (e.g., it describes incremental work), and it can significantly benefit from another round of revision. However, I won't object to accepting it if my co-reviewers champion it.

**Paper Topic And Main Contributions:**

This paper proposes an annotated resource to investigate to what extent LMs are able to account for theory of mind. Theory of Mind (ToM) is a theory that tries to explain one's own and others' mental states; it is at the heart of many cognitive processes, including language understanding.
Collected data also contain information on inferential steps as these are performed in the frame higher-order ToM: higher-order ToM involves recursively reasoning on others' beliefs, and is thus a major issue for the sort of knowledge globally grasped by LMs. Experiments were conducted on the newly released resource, employing two prompting approaches: Vanilla Prompting (VP) and Chain-of-Thought Prompting (CoTP); results were computed based on two metrics, standard and joint accuracy. In the latter setting, an answer was scored as correct only if all lower-order related questions within the same story were correctly answered. The GPT-4 and GPT-3.5 LMs were tested, showing a drop in performance on higher-order ToM tasks. The main results based on the analysis of results were as follows:
- CoTP prompting yields insignificant performance gains;
- Increased ToM order leads to decreased performances;
- LLMs' performance decreases as there are more deception communications involved.

I have a major concern, and I'd solicit Authors to elaborate on this point. LM are basically probability distributions over text sequences: as such, LMs are unfit to deal with logical reasoning or matters such as acquiring any sort of propositional knowledge (what is needed to face ToM related tasks).
In this sense, one is not surprised for the poor perfomance attained in the evaluation. Authors should then explain how they expect that whatever LM trained for dealing with ToM may generalize to previously unseen stories. How a device such as a LM is expected to be helpful in higher-order tasks, such as deciding whether 'Where does A4 think A3 thinks A2 thinks A1 thinks object O is?'?
Additionally, putting at the heart of the system a closed resource such as GPT* may be detrimental to investigate inner mechanisms and implementation details. Are the Authors willing to elaborate on these issues?
Such issues should be either addressed in the introductory Section, or discussed after providing the experimental results.


**Questions For The Authors:**

[Q1] I have a major concern, and I'd solicit Authors to elaborate on this point. LM are basically probability distributions over text sequences: as such, LMs are unfit to deal with logical reasoning or matters such as acquiring any sort of propositional knowledge (what is needed to face ToM related tasks).
In this sense, one is not surprised for the poor perfomance attained in the evaluation. Authors should then explain how they expect that whatever LM trained for dealing with ToM may generalize to previously unseen stories. How a device such as a LM is expected to be helpful in higher-order tasks, such as deciding whether 'Where does A4 think A3 thinks A2 thinks A1 thinks object O is?'?
Additionally, putting at the heart of the system a closed resource such as GPT* may be detrimental to investigate inner mechanisms and implementation details. Are the Authors willing to elaborate on these issues?
Such issues should be either addressed in the introductory Section, or discussed after providing the experimental results.

**Reasons To Accept:**

- The paper proposes an interesting work on a relevant cognitive device, such as Theory of Mind. This is a relevant theme to NLP, since it lies at the base of many cognitive processes, such as those involved in NL understanding.


**Reasons To Reject:**

- The computational approaches employed in the experiments seem insufficient to deal with the task at hand: or, equivalently, the task might not be appropriate for Language Models.

**Reproducibility:**

4: Could mostly reproduce the results, but there may be some variation because of sample variance or minor variations in their interpretation of the protocol or method.

**Reviewer Confidence:**

4: Quite sure. I tried to check the important points carefully. It's unlikely, though conceivable, that I missed something that should affect my ratings.

**Typos Grammar Style And Presentation Improvements:**

- line 189: "we incorporate distractor sentences that **relates** an agent" -> 'relate'

---

> ### Author Rebuttal · Authors · 2023-08-29
>
> Thank you for your thoughtful feedback. We appreciate the time you took to review our work and offer constructive comments. Below, we address each of your concerns:
>
> -   **Response on LMs’ incompatibility on ToM tasks:**
>     We will add a discussion on this point after the experimental results as follows:
> 	-   We acknowledge the view that LLMs, primarily being probability distributions over text sequences, might not excel in higher-order ToM tasks. However, several recent papers claimed that ToM has spontaneously emerged in LLMs as a byproduct of their development, or LLMs have acquired genuine ToM ability (Kosinski, 2023; Bubeck et al., 2023). Our findings challenge such a claim and expose the deficiency of current SOTA LLMs, especially for higher-order ToM.
> 	-   Still, we want to seek methods to equip LLMs with higher-order ToM ability, as it may help the model better deal with complicated scenarios in downstream tasks like emotional support and strategy advising. A recent paper [Sclar et al., 2023](https://arxiv.org/abs/2306.00924) introduces a new plug-and-play prompting method, SYMBOLICTOM, to endow LMs with a dramatically enhanced ability to reason about higher-order ToM. Thus, we believe that when equipped with proper modules, LLMs have the potential to complete ToM tasks, and this proposed method is one potential way. We will evaluate LLMs on our dataset with this approach in our follow-up revision.
>
> -   **Response on testing closed-source models:**
> We will add a discussion on this point after the experimental results as follows:
> 	- We concur with concerns about the opacity of closed-source models like GPT*. The detrimental effects of using black-boxed LLMs include unknown pre-training corpus, uncertainty about the reasons behind the model's errors, and difficulties in improving models’ performance (Ye et al., 2022; Ignat et al., 2023; Rodgers, 2023).
> 	- Therefore, in our paper, we also tested a competitive open-sourced LLM – Guanaco 65 B, which was the SOTA for the open-sourced ones when we wrote our paper. We believe that open-source models offer a clearer pathway to enhancing the logical reasoning capabilities of LMs (Rodgers, 2023). We advocate for increased community involvement in benchmarking these models.
>
> We hope our responses address your concerns and provide clarity on our methodology and findings. We are open to further suggestions to improve our manuscript.
>
>
> **Reference**
>
> Bubeck, Sébastien, et al. "Sparks of artificial general intelligence: Early experiments with gpt-4." arXiv preprint arXiv:2303.12712 (2023).
>
> Kosinski, Michal. "Theory of mind may have spontaneously emerged in large language models." arXiv preprint arXiv:2302.02083 (2023).
>
> Ignat, Oana, et al. "A PhD Student's Perspective on Research in NLP in the Era of Very Large Language Models." arXiv preprint arXiv:2305.12544 (2023).
>
> Rodgers, Anna. "Closed AI Models Make Bad Baselines." Towards Data Science, 2023. Accessed 23 May 2023.
>
> Sclar, Melanie, et al. "Minding Language Models'(Lack of) Theory of Mind: A Plug-and-Play Multi-Character Belief Tracker." arXiv preprint arXiv:2306.00924 (2023).

---

### Official Review · Reviewer_YSgP · 2023-08-05

**Soundness:** 4

**Excitement:**

3: Ambivalent: It has merits (e.g., it reports state-of-the-art results, the idea is nice), but there are key weaknesses (e.g., it describes incremental work), and it can significantly benefit from another round of revision. However, I won't object to accepting it if my co-reviewers champion it.

**Paper Topic And Main Contributions:**

This paper contributes a new dataset for measuring higher-order theory of mind capabilities in NLP models. Existing datasets in this domain are synthetically constructed and measure reasoning in stories up to 2nd order ToM. This paper takes their framework and introduces new templates to extend it to up to 4th order while introducing distractor sentences as well as previous work.

They compare the performances of LLMs like GPT-4, Claude, and Guanaco on this task using zero-shot simple and chain-of-thought prompting and show these models show a decreasing trend of performance as the order of ToM is increased showing the difficulty of this task. The authors show interesting discussions and analyses of model behaviors both when the models are right and when they are wrong.

In the end, the authors also discuss the implications of these results on NLP.

**Questions For The Authors:**

See weaknesses

**Reasons To Accept:**

1. An interesting dataset that is easy for people to create themselves (given the code with be released) and shows where current power LLMs still lag.
2. The analysis of model behaviors is very interesting and informative. They give insights into both the kinds of examples future datasets should contain and how new approaches can try to address the shortcomings of the current models and improve performance on such tasks.

**Reasons To Reject:**

1. The authors only evaluate the LLMs on zero-shot settings, all the models considered in this paper are capable of in-context or few-shot learning, these settings should be considered before claiming that the models perform poorly on this dataset. 3-4 step-by-step reasoning can easily be hand-written for these experiments.
2. I understand that theory of mind is an important reasoning skill that models should have. But after reading this paper, I am not super convinced that higher than the second-order theory of mind is a necessary reasoning ability in generalist language models (but maybe for those designed for specific tasks like chess or negotiation). Also, the examples in the dataset, at least the ones shown in the paper, are not very realistic. I am not arguing that this ability should not exist in LMs but I think a convincing realistic example should be provided to motivate the construction of this dataset in the introduction even though the examples in the dataset are synthetic.
3. Not a weakness per se, but I am not sure section 6 is adding much to the overall point of this paper. It seems like an extension of the introduction motivating why ToM is a necessary property to have. In my opinion, it should be absorbed in the introduction itself. On that note, this paper could easily be a short paper on some reorganization.
4. Finally, again not a criticism, but there has been some recent work on improving LMs' theory of mind capabilities which claim to work on any order of ToM: https://arxiv.org/abs/2306.00924. Not a reason to reject since this work just came out, but in a future version, the authors should evaluate their models with this approach.

**Reproducibility:**

5: Could easily reproduce the results.

**Reviewer Confidence:**

4: Quite sure. I tried to check the important points carefully. It's unlikely, though conceivable, that I missed something that should affect my ratings.

---

> ### Author Rebuttal · Authors · 2023-08-29
>
> Thank you for your valuable review. We are glad that you found our dataset interesting, and the analysis informative and insightful. Below, we address each of your concerns:
>
>
> * **Response on the zero-shot setting (weakness 1):**
> Regarding evaluations on only zero-shot settings, our primary objective is to assess the innate Theory of Mind (ToM) ability in LLMs. We believe that an LM with genuine ToM ability could master ToM tasks even under a zero-shot setting, as discussed in Shapira et al. (2023).  In the few-shot setting, the provided examples may bias the LLMs, and the models may end up simply imitating instead of genuinely conducting ToM reasoning. Such issues have been mentioned in prior work (Jin et al., 2022). Following previous research (Jin et al., 2022; Shapira et al., 2023), we also use a zero-shot setting to test the innate ability of LLMs. We will clarify this point in the revision.
>
>
> * **Response on the necessity of high-order ToM (weakness 2):**
> We appreciate the feedback. To strengthen the motivation of our dataset, we will include the following additional arguments in the paper:
> 	* One of the motivations to construct a higher-order ToM is that several recent papers claimed that LLMs acquire the ToM ability (Kosinski, 2023; Bubeck et al., 2023). As Liddle and Daniel (2021) show that human beings are able to conduct higher-order ToM ability, we want to check if LLMs indeed possess the genuine ToM ability like human beings as claimed by those papers.
> 	* From an application perspective, we believe that higher-order ToM ability is crucial for generalist LLMs. It enables better NLP applications for social goods such as emotional support and empathetic communication (Mitchell et al., 2015). For instance, in an emotional support scenario where the user seeks advice on dealing with multiple friends or families, LLMs need to first comprehend the complicated scenarios and the multi-layer thoughts of the people involved. Such applications require higher-order ToM to recursively reason about the feelings or thoughts among the multiple people involved.  In other use cases where LLMs need to consider the beliefs of multiple agents, such as telling a spy story, advising game strategy, and planning business or legal strategy, a generalist LM also utilizes its higher-order ToM ability.
>
> * **Response on the unrealistic introductory example (weakness 2):**
> We agree that we should add a concrete real-world example to motivate the construction of our dataset. We will modify Figure 1 in the revision as follows: We may use the scene shot from the famous TV series Friends: Phoebe says “They don't know that we know they know we know!” when she and Rachel try to identify whether Monica and Chandler know that Rachel and Phoebe know they are dating each other. Here is the [link](https://www.youtube.com/watch?v=wFmTEKnn64s) to the video clip in Friends, where the ToM increases from zeroth order to fourth order.
>
> * **Response on the paper structure:**
> We acknowledge the overlap in content and we will reduce some discussions in Section 6 on the importance of ToM ability and the applications of ToM in LLMs.
>
> * **Response on incorporating the recent works:**
> We're grateful for the reference to the recent work on LLMs' ToM capabilities. We will evaluate the models with this approach in our follow-up revision.
>
> **Reference**
>
> Bubeck, Sébastien, et al. "Sparks of artificial general intelligence: Early experiments with gpt-4." arXiv preprint arXiv:2303.12712 (2023).
>
> Jin, Zhijing, et al. "When to make exceptions: Exploring language models as accounts of human moral judgment." Advances in neural information processing systems 35 (2022): 28458-28473.
>
> Kosinski, Michal. "Theory of mind may have spontaneously emerged in large language models." arXiv preprint arXiv:2302.02083 (2023).
>
> Liddle, Bethany, and Daniel Nettle. "Higher-order theory of mind and social competence in school-age children." Journal of Cultural and Evolutionary Psychology 4.3-4 (2006): 231-244.
>
> Lucy, Li, and David Bamman. "Gender and representation bias in GPT-3 generated stories." Proceedings of the Third Workshop on Narrative Understanding. 2021.
>
> Malkin, Nikolay, et al. "GPT Perdetry Test: Generating new meanings for new words." Proceedings of the 2021 Conference of the North American Chapter of the Association for Computational Linguistics: Human Language Technologies. 2021.
>
> Mitchell, Rachel LC, and Louise H. Phillips. "The overlapping relationship between emotion perception and theory of mind." Neuropsychologia 70 (2015): 1-10.
>
> Shapira, Natalie, et al. "Clever hans or neural theory of mind? stress testing social reasoning in large language models." arXiv preprint arXiv:2305.14763 (2023).

---

### Meta-Review · Area_Chair_F9dU · 2023-09-15

**Recommendation:** 3

**Metareview:**

Reviewers provided scores 4,4,3 for soundness and 3,3,3 for excitement.

The following strengths and weaknesses were prominent in the reviews:

Strengths:
- addresses interesting cognitive device relevant to NLP (R2)
- interesting dataset that should be easy to use (R1, R3)
- analysis of model behavior (R1, R3)
- evaluate a range of LLMs (R3)

Weaknesses:

- only considers zero-shot settings (R1)
- questionable realism of examples (R1)
- it was questioned whether the task might be inappropriate for LLMs (R2, R3)
- dataset is small (R3) and implemention details are underspecified (R3)

---

### Decision · Program_Chairs · 2023-10-07

**Decision:**

Accept-Findings

**Comment:**

Reviewers provided scores 4,4,3 for soundness and 3,3,3 for excitement.

The following strengths and weaknesses were prominent in the reviews:

Strengths:
- addresses interesting cognitive device relevant to NLP (R2)
- interesting dataset that should be easy to use (R1, R3)
- analysis of model behavior (R1, R3)
- evaluate a range of LLMs (R3)

Weaknesses:

- only considers zero-shot settings (R1)
- questionable realism of examples (R1)
- it was questioned whether the task might be inappropriate for LLMs (R2, R3)
- dataset is small (R3) and implemention details are underspecified (R3)